

# Characterization of a modified printed optical particle spectrometer for high-frequency and high-precision laboratory and field measurements

Sabin Kasparoglu[1], Mohammad M. Islam[1], Nicholas Meskhidze[1], and Markus D. Petters[1]

[1]Department of Marine, Earth, and Atmospheric Sciences, NC State University, Raleigh, NC, 27695-8208, U.S.A

*Correspondence to*: Markus D. Petters (mdpetter@ncsu.edu)

**Abstract.** The Printed Optical Particle Spectrometer (POPS) is a lightweight, low-cost instrument for measurements of aerosol number concentrations and size distributions. This work reports on modifications of the Handix Scientific commercial version of the POPS to facilitate its use in multi-instrument aerosol sampling systems. The flow system is modified by replacing the internal pump with a needle valve and a vacuum pump. The instrument is integrated into closed-flow systems by routing the sheath flow from filtered inlet air. A high-precision multichannel analyzer (MCA) card is added to sample the analog pulse signal. The MCA card is polled at 10 Hz frequency using an external data acquisition system and improves upon the count-rate limitation associated with the POPS internal data acquisition system. The 90/10 rise and fall times for 10 Hz POPS data were measured to be 0.17 s and 0.41 s at a flow rate of 5 $cm^{-3}$ $s^{-1}$. This yields a sampling frequency of ~1-2 Hz below which the amplitude of measured fluctuations is captured with > 70 % efficiency. The modified POPS was integrated into the dual tandem differential mobility analyzer system, to explore the coalescence of dimer particles. Results show that the pulse-height response increases upon dimer coalescence. The magnitude of the increase is broadly consistent with the change in light scattering amplitude predicted by the T-matrix method. It is anticipated that this modified version of the POPS will extend the utilization of the technique for a range of field and laboratory applications.

**Keywords:** optical particle spectrometer, aerosol measurement, aerosol viscosity

**Short summary:** A modified version of a Handix Scientific's Printed Optical Particle Spectrometer is introduced. The manuscript presents characterization experiments, including concentration-, size-, and time-responses. Integration of an external multichannel analyzer card removes counting limitations of the original instrument. It is shown that the high-resolution light-scattering amplitude data can be used to sense particle phase transitions.



## 1. Introduction

Aerosol particles in the atmosphere are important as they impact climate and human health (Pöschl, 2005). Measurements of the aerosol number concentration and size distribution are critical to understand their effects on human health and the environment (Flagan, 2014) and to help devise strategies that control atmospheric particle population (Väkevä et al., 2000). Optical particle counters (OPCs) have a long history in aerosol measurement. In OPCs, aerosol flow is passed through a laser beam. The scattered light is collected by a detector and recorded as a voltage pulse (Hinds, 1999). The amplitude of the scattered light signal depends on the size, refractive index, and shape of the particle (Quinten et al., 2001). Modern OPCs are widely used in a variety of applications including measuring the size distribution of ambient aerosols (Burkart et al., 2010; Snider and Petters, 2008), size distributions of ambient aerosols coupled with differential mobility analyzers (DMA) (Stolzenburg et al., 1998), the size of hygroscopically grown particles in humidified aerosols (Sorooshian et al., 2008; Wex et al., 2006), or number concentration of droplets and ice nucleation particles exiting ice nucleation instrumentation (Kulkarni et al., 2020; Möhler et al., 2020; Wolf et al., 2020). In conjunction with other instruments, the amplitude of the light scattering signal can also be used to measure aerosol refractive index (Hand and Kreidenweis, 2002; Kiselev et al., 2005; Wex et al., 2009).

Some commercially available OPCs can measure particles down to 60 nm in diameter (Cai et al., 2008; Moore et al., 2021). However, the lower size threshold varies between different instrument designs with ~100 nm and ~300 nm being more typical lower size cuts. Accessing lower sizes requires more powerful lasers and thus units with lower size cuts are generally more expensive. Gao et al. (2016) introduced a Printed Optical Particle Spectrometer (POPS) to measure aerosol sizes between ~140 and ~3000 nm. A commercial version of this instrument is now available from Handix Scientific (1613 Prospect Park Way, Suite 100, Fort Collins, Colorado 80525 USA). The POPS is attractive for its relatively low cost, lightweight, low power consumption, and its ability to access sizes smaller than 200 nm. It's been used in previous studies to obtain vertical profile size distribution on unmanned systems (de Boer et al., 2016, 2018; Creamean et al., 2021; Kezoudi et al., 2021; Mascaut et al., 2022; Telg et al., 2017; Yu et al., 2017), indoor particle size distributions (Boedicker et al., 2021), and other real-time aerosol number size distributions measured during field campaigns (Brock et al., 2019; Liu et al., 2021; Ranjithkumar et al., 2021; Zhang et al., 2019).

The relatively low cost of POPS introduces some limitations. Data acquisition is performed via a single board computer, which processes a conditioned analog signal from the photomultiplier tube via a digitizer. This hardware is limited by a maximum count rate that makes the instrument unsuitable for measurements with number concentrations exceeding ~2,000 cm$^{-3}$. The commercial version of the instrument saves the data on a MicroSD card. The stored data is in a binary format and can be configured to save single-particle information, including precise timestamps for each beam transit. This, however, makes integration of the POPS into a system with multiple sensors challenging, as time stamps must be kept synchronized across multiple computers. Having a precise timestamp is especially critical in systems that rely on the precise timing of the signals, e.g., for eddy covariance particle flux measurements (Emerson et al., 2018; Farmer et al., 2011). The



POPS data acquisition software also provides binned data into a user-selectable number of channels. The default is 16 bins. This data can be streamed to an external computer at 1 Hz time resolution via a standard serial connection. While this allows for integration with other systems, the serial data stream yields data with degraded time and bin resolution. Finally, the instrument is designed as a low-power and light-weight system with a miniature rotary vane pump (Gardner Denver Thomas GmbH, Germany, Model G 6/01-K-LCL) regulating the sample flow. This also makes physical integration of the instrument into a closed flow path challenging, especially when there is an appreciable pressure drop in the system, e.g. for tandem DMA-OPC experiments (Park et al., 2008; Sorooshian et al., 2008) .

To alleviate these limitations, we modified the commercial version of the POPS by adding a high-resolution external multichannel analyzer (MCA) card and modified the flow path. We present characterization experiments of the modified POPS instrument, including concentration-, size-, and time-responses. In this study, we also demonstrate a new application of the instrument that uses the light-scattering amplitude to detect particle phase transitions when integrating it into a multi-DMA setup.

## 2. Methods

### 2.1 Modified POPS

The schematic of the modified POPS is shown in Fig. 1. An MCA card (AMPTEK Inc., 14 DeAngelo Drive, Bedford, MA, USA, Model MCA 8000D PA) is interfaced with the analog output of the photomultiplier electronics board. A 50 Ω terminator was integrated to reduce electronic noise. The MCA card consists of a preamplifier and a shaping amplifier that produces peak amplitudes proportional to analog pulse height. The card allows configuration for a high gain (0-1 V) and low gain (0-10 V) setting. In this work, the high gain setting was used. The card's electronics generate a calibrated 8191 channel pulse-height count distribution, which is polled at 10 Hz frequency into a custom external data acquisition system. We refer to these data as MCA-PH distributions. Integration over the MCA-PH count distribution and accounting for flow rate yields the MCA derived number concentration, hereafter denoted "MCA concentration". This system simultaneously records data from the serial stream provided by the POPS internal data acquisition system at 1 Hz resolution. These data provide a time series of measured sample flow rate and manufacturer binned count histograms of the size distribution at 16 bin resolution. Integration over the count size distribution histogram yields the serial signal derived number concentration, hereafter referred to as "Serial concentration". In addition, the POPS' own computer digitizes the photomultiplier tube (PMT) signal and extracts peak information on a particle-by-particle basis. The instrument was configured to save single-particle data onto an external disk in binary format. After completion of an experiment, the data is extracted to create pulse-height histograms, size distributions, and concentration time series. The pulse height histograms are binned into 65535 channels, corresponding to the maximum resolution of the 16-bit digitizer card used in the instrument (Gao et al., 2016). We refer to these data as Digitizer-PH distributions. Integration over the Digitizer-PH count distribution and accounting for flow rate yields to



digitizer-derived number concentration, hereafter denoted as "Digitizer concentration". The three data streams are merged based on the continually monitored time offset between the two computers' clocks.

In the original instrument design, the sheath flow is taken from the ambient air. Here it is split from a total inlet to
facilitate integration into complex flow systems. The original miniature pump is replaced with a needle valve and vacuum pump, operated at critical flow conditions. This provides flow stability when operating downstream of other instruments. The needle valve is adjusted to achieve the desired sample flow, which is measured using a laminar flow element. The sheath flow rate is not actively controlled. At a sample flow rate of 0.3 L min$^{-1}$ (5 cm$^{-3}$ s$^{-1}$), the total flow is ~0.9 L min$^{-1}$. All tests were performed at a sample flow rate of 0.3 L min$^{-1}$.

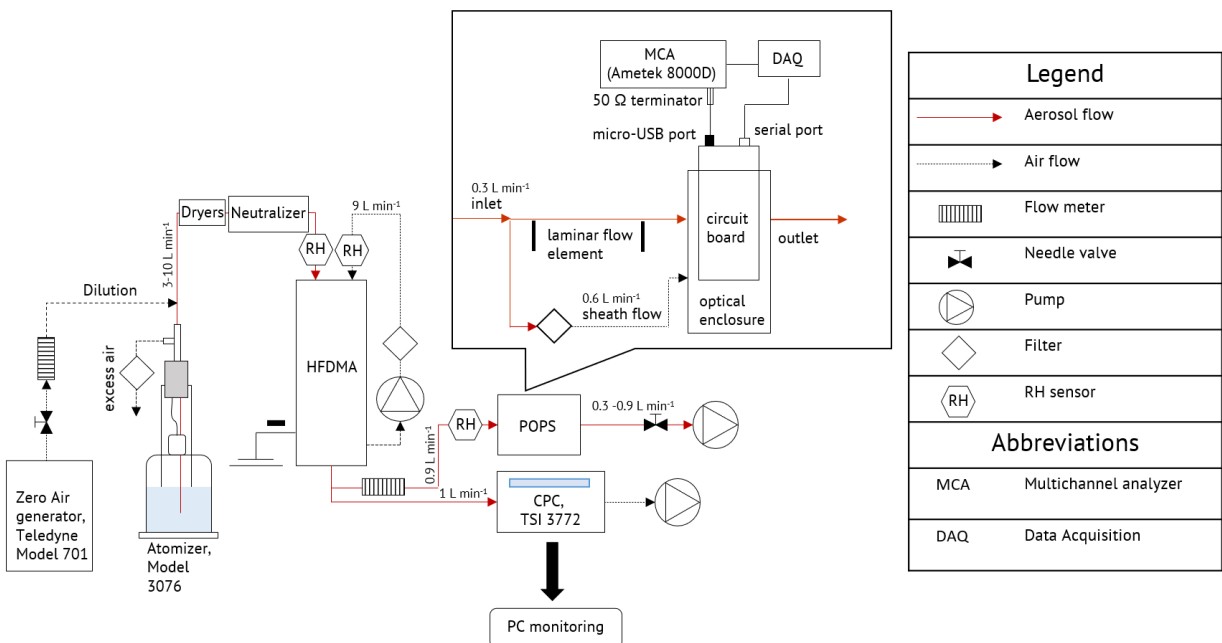


**Figure 1: Schematic of the modified POPS instrument and size-resolved characterization experiments.**

**2.2 Concentration and Size Response**

The response of the Digitizer-PH and MCA-PH data were characterized downstream of a custom DMA system (see the schematic in Fig. 2). Particles were taken either from laboratory air or from atomized solutions (TSI Inc. Model 3076,
Shoreview, MN, USA). Atomized particles were dried to < 25% RH using 4 or 5 silica-gel driers in series. Particles were neutralized using either [210]Po or an x-ray neutralizer (TSI Inc. Model 3088). The DMA consisted of a high-flow column (Stolzenburg et al., 1998) operated at 9 L min$^{-1}$ sheath flow. A condensation particle counter (TSI Inc. Model 3772) was placed in parallel to the POPS to measure the particle concentration (see Fig. 1). Concentration from the condensation particle counter (CPC) was taken from the serial data stream which includes a coincidence correction by the manufacturer.
The MCA concentration was corrected for coincidence using the method by Collins et al. (2013) and assuming a beam



transit time of 5 µs. No coincidence correction was performed for the Digitizer concentration since this data stream produces a concentration error that is primarily due to electronic limitations.

For size response characterization, the number concentrations of particles visible to the POPS was controlled to be <1,000 cm$^{-3}$ to avoid saturation of the digitizer response signal (Gao et al., 2016; Mei et al., 2020). The bin resolution of the
Digitizer-PH and MCA-PH was reduced to ~500 bins to improve counting statistics and to present the data at the same bin resolution. For each size, counts were accumulated over 2 min and the resulting pulse-height count histogram was normalized by the maximum value of the histogram.

## 2.3 Time response characterization

The response of the modified instrument to the instantaneous change in particle number concentration describes the time
response (Enroth et al., 2018). Here, the time response was measured by placing a 3-way solenoid valve switching between filtered and laboratory air upstream of the POPS inlet. The 90/10 rise and fall time describes the time required for the concentration to change between 10 % and 90 % and 90 % and 10 % of the value corresponding to a toggle between the ambient air and the filter. Using a resistor–capacitor (RC) circuit as analogue, the 3 dB cut-off frequency can be obtained as 0.35/tr where tr is the 90/10 rise or fall time in the output signal for the RC low pass filter system (Andrews, 1999). The
90/10 rise and fall times were determined for the 10 Hz MCA concentration and 1 Hz Serial concentration data streams.

## 2.4 Phase state response

Utilization of the Digitizer-PH data for measuring aerosol phase transitions was tested by integrating the POPS into a dual tandem DMA setup (Kasparoglu et al., 2021; Rothfuss et al., 2019; Rothfuss and Petters, 2016). Briefly, the method uses two DMAs to select particles with the same mobility but using different polarities. The streams are merged into a coagulation
chamber. Coagulation events of +1/−1 and +2/−2 charged particles result in charge-neutral dimer particles. Initially these dimer particles consist of touching spheres, and we also refer to those as bispheres. Particles exiting the coagulation chamber are passed through an electrostatic precipitator (Kasparoglu et al., 2022) to remove all charged particles. Next, particles are passed through a coalescence chamber which exposes the particles for a fixed time to elevated temperature conditions. If the temperature is sufficiently high, the particles coalesce and transform from a touching bisphere dimer particle into a spherical
particle. The sphere equivalent diameter of a dimer particle after coalescence is $\sqrt[3]{2}D_{mono}$, where $D_{mono}$ is the monomer diameter from which the dimer particle has been formed. The transformation from uncoalesced dimer particles to spherical particles is a change in the particle morphology. The change in morphology is monitored by measuring the mobility diameter using a scanning mobility particle sizer (SMPS) downstream of the coalescence chamber. Here, the objective is to test if the transition from the bisphere to the sphere can be sensed using the modified POPS.

Figure 2 provides a high-level overview of the setup. The detailed setup of the system is described in previous publications (Kasparoglu et al., 2021; Rothfuss and Petters, 2016; S.Petters et al., 2019). Dry and charge-neutralized particles were generated via atomization as described earlier. The upstream DMAs consisted of a high-flow DMA (HFDMA)



column (Stolzenburg et al., 1998) that was operated at 9 L min$^{-1}$ flow and with and a negative high-voltage power supply and a TSI 3071 column (TSI DMA) that was operated at 5 L min$^{-1}$ sheath flow and with a positive high-voltage power supply.

The coagulation chamber consisted of 50 feet (15.24 m) long ⅝" (1.59 cm) outer diameter copper coil. The coalescence chamber consisted of a ~22 cm$^3$ heat exchanger mounted to a thermoelectric heat exchanger. The flow rates through the coagulation and coalescence chamber were 0.5 L min$^{-1}$. The SMPS downstream of the coalescence chamber consisted of a TSI 3071 column operated at a 3 L min$^{-1}$ flow rate. In the original version of the instrument, the particle concentration downstream of the SMPS is monitored using a CPC. Here, that CPC was replaced with the modified POPS.

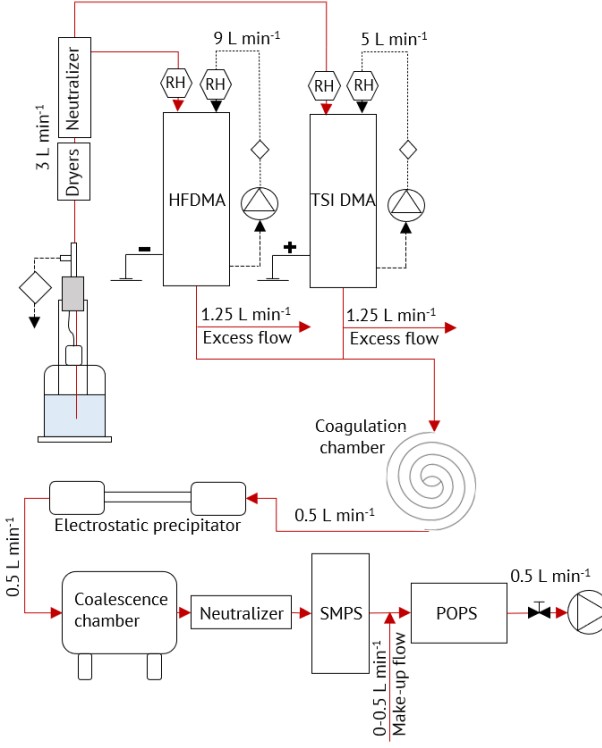


**Figure 2: Schematic of the modified POPS instrument integrated into a dual tandem DMA setup. Symbols are as in Fig. 1.**

The standard processing of phase transition data from the dual tandem DMA technique is described in detail in previous studies (Rothfuss and Petters, 2016, Kasparoglu et al. 2021, 2022). The mobility distribution is determined by aligning the voltage applied to the SMPS and the number concentration measured by the CPC. The same analysis can be

performed by treating the POPS as just a particle counter. The change in the mode mobility diameter is tracked as a function of temperature in the coalescence chamber. The mode diameter decreases when the particle transitions from an uncoalsesced dimer particle to a sphere. This is because the drag force acting on the sphere equivalent particle is less than the drag on the uncoalesced dimer particle. The resulting data of mode mobility diameter versus temperature forms a sigmoid curve, with smaller mode diameters occurring at larger temperature. The midpoint of the transition defines the transition temperature and



is determined from a fit to the data. The transition temperature corresponds to the condition when viscosity is ~$10^7$ Pa s, which is determined from sintering theory (Rothfuss and Petters, 2016).

In addition to the number concentration data, the POPS also measures the PH data. To track the PH data across the transition from uncoalesced dimer particle to coalesced sphere, the Digitizer-PH was determined at ~±4% of the mode diameter. The range was selected to obtain enough counts to construct the PH histogram and to determine the mean PH as a
function of temperature. This allows for tracking the light-scattering amplitude as the particle undergoes the transition from uncoalesced dimer to coalesced sphere. As we will show next, the light-scattering amplitude is predicted to increase for this transition for dimer particles formed from < 300 nm diameter monomers.

## 2.5 Light scattering calculations

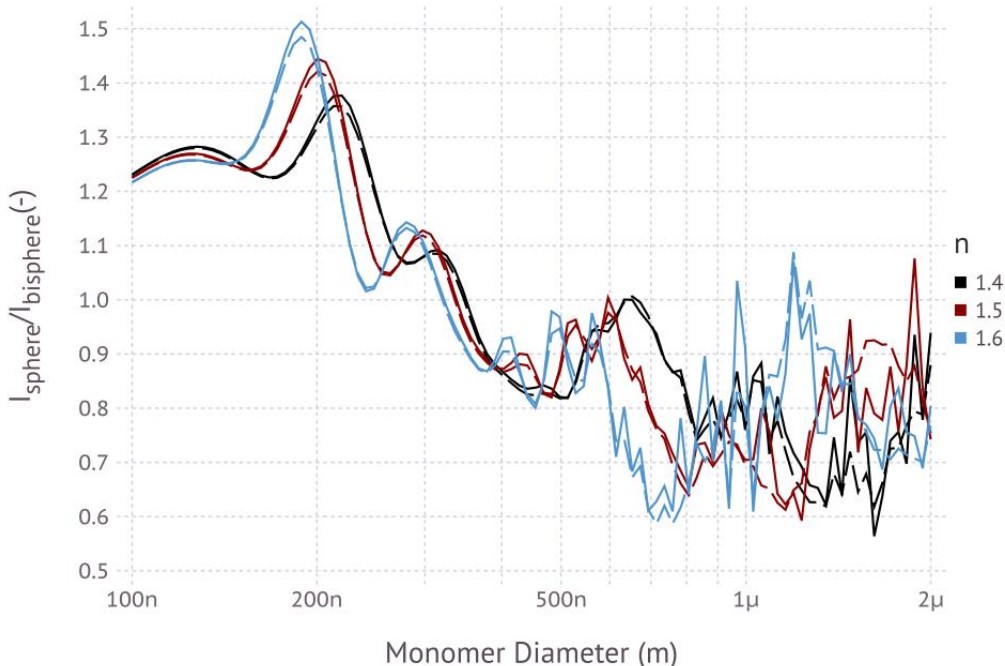

**Figure 3: The intensity ratio of the uncoalesced bisphere particle and equivalent spherical particle as a function of monomer diameter and refractive index. Each line corresponds to a refractive index $m = n + ki$, where $n$ is the real and $k$ is the imaginary part. Colors correspond to variations in $n$. Solid lines correspond to $k = 0$ and dashed lines correspond $k = 0.01$.**

Light scattering amplitudes of spheres and randomly oriented dimer particles were computed using the method and Fortran
code developed by Mackowski and Mishenko (Mackowski, 1994; Mishchenko, 1991; Mishchenko and Mackowski, 1994). The computational method is based on the superposition T-matrix approach. The code predicts the scattering phase function for bispherical particles with diameters $D_1$ and $D_2$. The distance between sphere centers was set such that the spheres are touching. We confirmed that in the limit $D_2 = 0$ the algorithm converges to the phase function calculated from Mie theory.



The phase function for the scattered light intensity, $I$, was integrated from 38 to 142 degrees for a wavelength of 405 nm.
The scattering angles and wavelength correspond to the configuration of the POPS (Gao et al., 2016).

Figure 3 shows the calculated light intensity ratio of the uncoalesced bisphere particle and equivalent spherical particle as a function of monomer diameter and refractive index. Typically, organics have a refractive index of 1.4-1.5 (Flores et al., 2014; He et al., 2018; Nakayama et al., 2012), sea salt has a refractive index of 1.51 (Bi et al., 2018) , which are spanning the range of most atmospheric aerosol types (Mico et al., 2019; Shepherd et al., 2018). Some secondary organic
aerosol particles from monoterpenes such as limonene, a humulene, a pinene, have an imaginary part of the refractive index, $k < 0.05$ (Flores et al., 2014). On the other hand, strongly absorptive material, such as black carbon, has a $0.5 < k < 1$ imaginary part of the refractive index (Pluchino et al., 1980; Bond and Bergstrom, 2006; Janzen, 1979). For monomer particles $< \sim300$ nm, the intensity is predicted to increase upon coalescence. The magnitude of the increase weakly depends on the real and imaginary part of the refractive index, but the behavior is broadly consistent over the range of the assumed
optical properties shown in Fig. 3. For monomer sizes $> \sim 300$ nm, the intensity ratio generally decreases with the monomer diameter. However, the magnitude depends more strongly on the refractive index. As we will show later, the change in scattering intensity can be used to measure the change in particle morphology during particle coalescence.

## 3 Results

### 3.1 Size response

Figure 4 shows the response of the Digitizer-PH and MCA-PH distributions for the mobility of selected dried ammonium sulfate particles. The PH histograms resolve the output distribution of size-classified particles by a DMA. As expected, the mode of both the Digitizer-PH and MCA-PH histograms scale with the selected mobility diameter. The dependence of the PH response on refractive index and its scaling with Mie theory have been discussed elsewhere (Gao et al., 2016, Mei et al., 2020) and thus it is not further analyzed here. The 200 nm to 250 nm particles PH histograms shows a shoulder to the right,
corresponding to doubly charged particles transmitted by the DMA. Compared to the Digitizer-PH, the MCA-PH distribution does not fully resolve the main mode of the 200 nm particle distribution generated by the DMA. Furthermore, the multiply charged particle shoulder appears to be enhanced. However, the main mode is resolved for 225 nm particles. This suggests that the smallest detectable particle size is slightly larger than 200 nm for the MCA-PH mode of data acquisition. The data show that the MCA-PH and Digitizer-PH are strongly correlated for diameters greater than 250 nm.
Deviations for the smaller sizes are due to the drop in detection efficiency of the MCA-PH distribution. We attribute these to the treatment of noise and resolution limitations of the MCA card at low signal voltages.





**Figure 4: Top and middle panels: Digitizer-PH and MCA-PH histograms for dry size selected ammonium sulfate particles between 200 nm and 700 nm. The histograms were normalized such that the largest value equals unity. Bottom panel: comparison of Digitizer-PH and MCA-PH at the peak of the histogram. Error bars correspond to the 95% confidence interval of the mode determined via a curve fit of the data to a lognormal distribution. The line corresponds to linear regression with the equation Digitizer-PH=MCA-PH*50.59-92.57.**

### 3.2 Concentration response

Figure 5 shows the comparison between Digitizer concentration, MCA concentration, and CPC reference concentration. In these experiments, concentration was varied using a dilution valve and size-selected particles > 250 nm, corresponding to sizes where the MCA Concentration and Digitizer concentration agreed. As reported previously (Gao et al., 2016; Mei et al., 2020), the Digitizer concentration strongly deviates for concentrations exceeding 2,000 cm$^{-3}$ and the signal saturates below 5,000 cm$^{-3}$. In contrast, the coincidence corrected MCA concentration correlates well with the CPC concentration up to 15,000 cm$^{-3}$. Note that this particular CPC model reports values at concentrations > 10,000 cm$^{-3}$. However, per manufacturer



specification, the optimal performance range is expected to be achieved at < 10,000 cm$^{-3}$. Thus, extending the comparison to even higher concentrations was not possible. However, we expect that higher concentrations can be measured with the MCA-PH POPS without difficulty. The deadtime of the MCA card is only 10 ns. At a concentration of 100,000 cm$^{-3}$ and a flow rate of 5 cm$^3$ s$^{-1}$, the mean interarrival time is 50 µs, which is orders of magnitude larger than the deadtime. Thus, a deadtime correction is not needed and electronic counting limitations are unlikely.

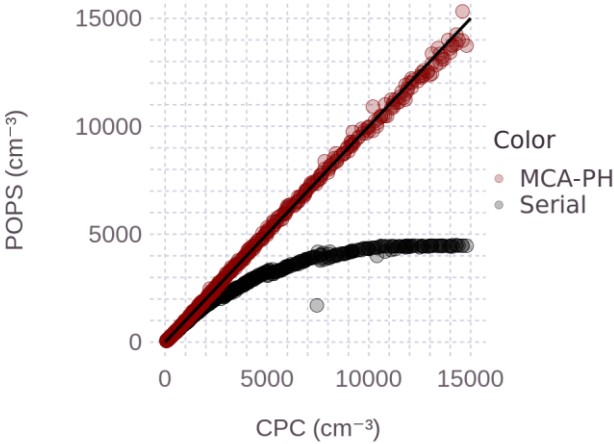


**Figure 5: Relationship between CPC number concentration and POPS number concentration derived from the Serial and MCA-PH distributions. The solid black line corresponds to the 1:1 relationship.**

**3.3 Time response**

Figure 6 shows the measured time response derived from the Serial concentration and MCA concentration. Note that the
concentration of the MCA-PH stream is lower than that of the Serial stream due to the difference in the minimum detected size between the digitizer and the MCA card (Fig. 4). The left panel in Fig. 6 shows multiple toggles between the instrument sampling filter and room air. The right panels in Fig. 6 show zoomed-in versions of a single scan. The rise and fall times averaged by four cycles are 0.17 s and 0.41 s for the 10 Hz MCA concentration and 0.81 s and 1.44 s for the 1 Hz Serial concentration. Multiple data points trace out the time response, especially during the decay. Thus, the actual time response of
the instrument is slower than the 10 Hz sample interval. This suggests that the measured response corresponds to the limit imposed by the flow geometry. We did not record the single-particle output for these experiments. However, as we will show next, 10 Hz binned Digitizer-PH derived concentrations have a similar time response as the 10 Hz MCA derived concentrations.





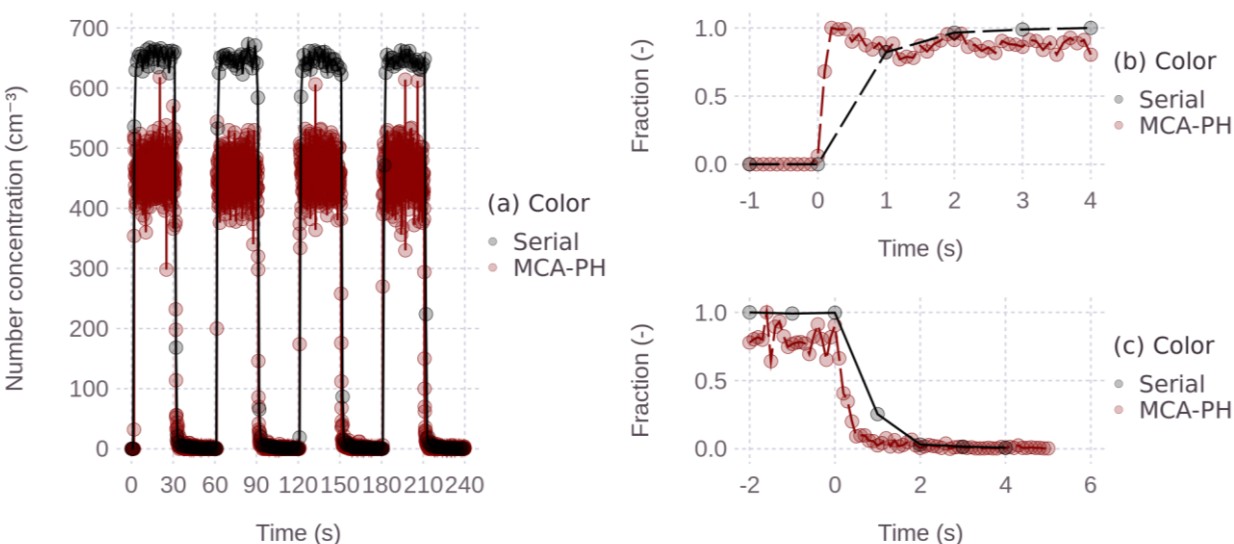

**Figure 6: Left: time response curves were measured by the POPS data acquisition system and MCA data acquisition system. Right: zoomed-in view for one example up (top) and down (bottom) cycle. Time $t = 0$ s corresponds to the valve switch.**

### 3.4 Ambient measurements

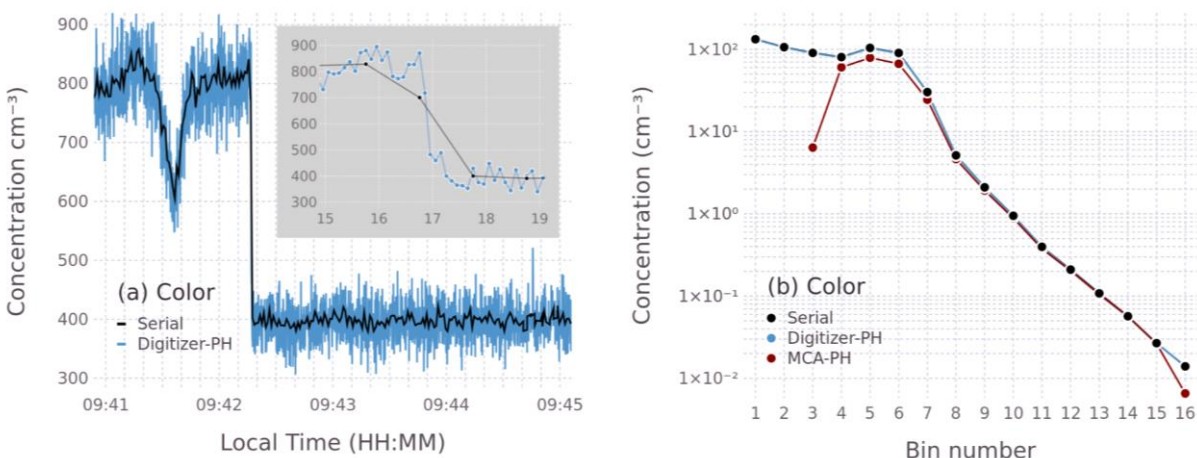

**Figure 7: (a) Comparison of 1 Hz Serial concentration and 10 Hz Digitizer concentration. The inset is a magnified view corresponding to the transition between 9:42:15 and 9:42:19. (b) Comparison of the Serial distribution, Digitizer-PH distribution, and MCA-PH distributions. Serial and Digitizer-PH distributions are indistinguishable.**

Figure 7 summarizes ambient measurements using air sampled from outside the laboratory and air inside the laboratory. The data show that the concentration time series and size distribution reported by the modified POPS' serial data stream can be reconstructed from Digitizer-PH data. While this closure should be expected, it shows that the Digitizer-PH data can be binned at a higher than the manufacturer selected frequency. At ~9:42:17, the sampling line from the outside was disconnected at the inlet of the POPS, thus simulating a step-change like the filter toggle test shown in Fig. 6. As shown in



the inset of panel (a), the rebinned 10 Hz Digitizer-PH concentration time response is equivalent to the 10 Hz MCA-PH response shown in Fig. 6. Reconstructed size distributions for the whole period are identical between the onboard serial processing and Digitizer-PH postprocessed distribution if the same binning algorithm is used. (The onboard binning

algorithm divides the Digitizer-PH distribution into 16 logarithmically spaced PH bins between a prescribed minimum and maximum PH.)

      The MCA-PH distribution compares reasonably well with the Digitizer-PH derived distribution. Here the MCA-PH distribution is obtained using the linear relationship between MCA-PH and Digitizer-PH shown in Fig. 4, followed by applying the Digitizer-PH binning algorithm to the calibrated MCA-PH signal. The MCA-PH derived distribution is in

excellent agreement for bins 8-15. There is a discrepancy in channel 16 which is due to the upper limit of the detectable MCA-PH at the high gain setting (0-1 V). Applying the relationship between the pulse height from Fig. 4 implies that the highest equivalent Digitizer-PH that can be measured by the MCA card at this setting is 50399. This contrasts with a maximum digitizer PH by the POPS hardware is 65535. Therefore, the MCA card is missing counts in bin 16 due to signal saturation. The detection efficiency is reduced by 10-30 % in bins 4-7, reduced by more than a factor of 10 in bin 3, and zero

in bins 1 and 2. This drop off in counting efficiency is consistent with the observed size response by the MCA-PH compared to the Digitizer-PH signal reported in Fig. 4.

### 3.5 Phase state detection

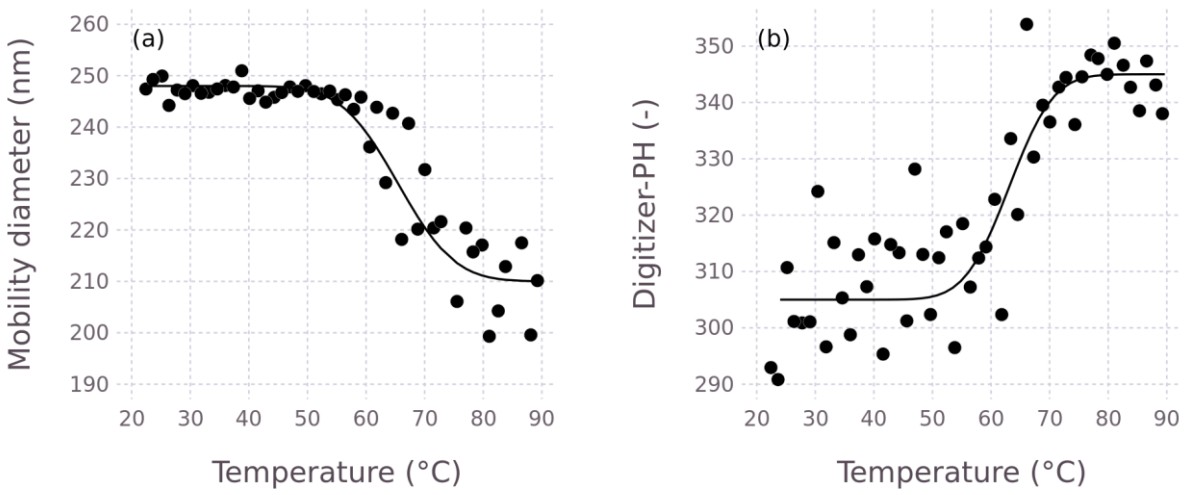

**Figure 8: The phase transition of bisphere organic particles between 20-90 °C. Black lines correspond to those of a fitted logistic**
**curve. (a) Phase transition of dimers for the nominal 166 nm glucose-sucrose particles versus temperature. The midpoint of the transition is $T_r \pm \sigma$ = 65.5±9.8 °C. (b) Temperature scans of Digitizer-PH for the nominal 166 nm glucose-sucrose particles versus temperature. The midpoint of the relaxation is $T_r \pm \sigma$ = 63.1±5.9 °C.**





Figure 8a shows the change in the mode diameter as a function of increasing the temperature in the coalescence chamber. At ~60 °C, the mode diameter starts decreasing with increasing temperature due to the onset of coalescence, which causes particles to become more spherical. At ~72 °C, all particles have fully coalesced and no further changes in electrical mobility diameter are observed. Figure 8b shows the measured Digitizer-PH at the peak of the mobility distribution. At ~60 °C, the scattered light intensity starts to increase. The increase stops at ~72 °C when all coalescence is complete. The observed increase is 13% in scattering intensity, derived from an increase from 305 to 345 in Digitizer units. Note that the real part of the refractive index of glucose and sucrose at 405 nm are ~1.35 (Bodurov et al., 2017), compared to 1.54 for ammonium sulfate used in Figure 4 (Washenfelder et al., 2013). The observed increase is less than the predicted value of ~20 % by the T-matrix code, possibly due to a combination of measurement uncertainties and slightly irregular shape of the monomers from which the dimer particles are generated. Nevertheless, the data clearly show that the coalescence leads to an increase in scattered light intensity. Thus, this experiment demonstrates how the high-resolution pulse height data of the modified POPS can be used in laboratory-based research.

## 4 Discussion and Conclusions

We modified the commercial version of the POPS to add an MCA card for data acquisition and to modify the flow path to facilitate integration into complex flow systems. The data presented here show that both the pulse height data created by the MCA card and single particle data stored by the onboard computer of the POPS can be used to acquire count histograms at 10 Hz time resolution, while also preserving the native 13 bit and 16 bit bin resolution of the MCA and digitizer hardware. A 10 Hz (100 ms) sampling frequency exceeds the physical time response of the system. The effective measured 10/90- and 90/10-time response is 170 and 410 ms. Using a RC circuit model (Andrews, 1999) and applying the estimate 0.35/tr to determine the 3 dB cut-off frequency, we deduce that fluctuation can be captured with > 70 % efficiency at < ~1 Hz. However, this cannot be achieved with the POPS standard data output and requires sampling at frequencies > 1 Hz using either the MCA-PH data or postprocessing of the single-particle Digitizer-PH data at 10 Hz resolution.

For many applications, the Digitizer-PH data acquisition is superior to that of the MCA-PH data acquisition. The digitizer data has a better size resolution, resolves to smaller sizes, and has a larger dynamic size range when compared to the MCA data acquisition of the same signal. The Digitizer-PH data can be post processed at 10 Hz frequency while preserving the full resolution of the Digitizer-PH response. When combined with time-stamped serial data acquisition, the Digitizer-PH data can also be used in an integrated system whose precise timing relies on an external clock. One example is an eddy covariance system, where timestamps of the sonic anemometer and other sensors must remain tightly synchronized. Even though the Digitizer-PH has a better size resolution, the MCA mode of data acquisition can be useful for a range of applications. Full resolution 10 Hz data can be obtained on an external data acquisition system in real-time. Unlike the Digitizer, the MCA card allows reliable measurement of number concentrations for values exceeding 2,000 cm$^{-3}$, using standard methods of coincidence correction (Collins et al., 2013). This is particularly useful when using the instrument as a

detector behind a DMA when working with atomized particles, where concentrations may vary by orders of magnitude between 150 nm and 3 μm. Importantly, the addition of the MCA still preserves the Digitizer-PH output. Thus, the combination of the two may be used to reconstruct the size distribution from the Digitizer-PH using post-processing while scaling the concentration using the measured MCA-PH data. The combination of real-time processing and the ability to measure high-concentration at high time resolution could also enable the use of the modified POPS system for the

spatiotemporal characterization of high-concentration aerosols. For example, mobile measurements of open combustion near sources or the characterization of volatile electronic cigarette vapors that are not amenable to dilution without causing evaporation (Wright et al., 2016).

        As shown in Fig. 4, the full resolution pulse height histograms provide detailed information about the size-selected aerosol size distribution. While the utility of the high-resolution PH data is limited in ambient measurements due to unknown

refractive index, this data can be useful in laboratory research. One example shown here is the use of the POPS to detect phase transitions of bispheric particles to changes in optical properties. As particles transition from a bisphere to a sphere, the scattering signal increases within a certain size range. This increase is readily measured by the modified POPS. Based on our data we estimate that relative changes in scattering intensity of +/- 2-3 % can be resolved with the Digitizer-PH signal.

**Code and data availability**

Scripts and data used to generate the figures will be made available via zenodo.org upon final acceptance of the manuscript.

**Author contribution**

MDP conceptualized the study. NM and MDP were responsible for funding acquisition. SK and MMI performed the experiments and analyzed the data. MDP performed the T-matrix simulations. SK and MDP wrote the initial draft. All authors commented on the paper.

**Competing interests**

The authors declare that they have no conflict of interest.

**Acknowledgements**

This work was supported by grants DE-SC0018265 and DE-SC0021074. We thank Gavin McMeeking and Bryan Rainwater at Handix Scientific for their help with interpreting the raw POPS data.



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
