# Peer review of "Characterization of a modified printed optical particle spectrometer for high-frequency and high-precision laboratory and field measurements"

_Atmospheric Measurement Techniques, 2022_

## Author Comment (AC1)

**08/19/2022**

**RC1: 'Comment on amt-2022-185', Anonymous Referee #1, 16 Jul 2022**

Author statement: The authors thank the referee for reviewing this manuscript. An itemized **response** for the rebuttal can be found below for each response given in blue. The tracked-changes version can be found below for each response in *red* and the already existing text is in *italic*.

General comments:

This paper introduces a modified version of a Handix POPS and demonstrates the performance improvement in measuring the aerosol number concentration, particle size and signal time responses. The improvement is promising and very important to expand the current POPS capability. In addition, the authors presented a very useful application to monitor the particle phase transition with the high-resolution light scattering amplitude data. Thus, the reviewer recommends publication after addressing the below concerns.

1. What is the modified POPS's total weight, power requirement, and cost? What is the targeted application of the modified POPS? Is it maintained the current lightweight and low-cost sensor application?

   **Response 1:** The following is added to the manuscript.

**Manuscript:**

Section 2.1:

*The schematic of the modified POPS is shown in Fig. 1. A photograph of the modified unit is provided in Fig. 2.*

Discussion:

*We modified the commercial version of the POPS to add an MCA card for data acquisition and to modify the flow path to facilitate integration into complex flow systems. As shown in Fig. 2, the unit was placed inside a 19'' rackmount enclosure. No effort was made to minimize the height and weight of the enclosure or the power consumption of the unit. The additional power requirement beyond the factory design depends on the vacuum pump. We used an ~400W model, but smaller pumps will be sufficient. The rackmount form factor is suitable for laboratory and field applications, including measurements from mobile platforms such as vehicles or airplanes where size and weight are less critical than in balloon-borne deployments. The cost to modify the factory supplied unit was ~US $6,500, including the MCA card, enclosure, vacuum pump, needle valve, power supply, Swagelok fittings, electrical connections, and particle filters.* The data presented here show that both the pulse height …

1: MCA card (Amptek 8000D)
2: Power supply (LRS 75 12 Mean well)
3: POPS
4: 50 Ω terminator
5: Needle valve
6: Data acquisition board
7: Inlet flow
8: Vacuum pump line
9: Serial cable/serial port

***Figure 2:*** *Photograph of the modified POPS.*

When the modified POPS was integrated into a dual tandem DMA, how were the multiple charges treated in the data reduction?

> **Response:** The following was added to the manuscript

*Manuscript:* *To track the PH data across the transition from uncoalesced dimer particle to coalesced sphere, the Digitizer-PH was determined at ~±4% of the mode diameter* *of the dimer* *peak. Note that the mobility distribution of the dual tandem DMA is complex and contains multiple modes. Details are given in Petters (2018) and Rothfuss et al. (2019). The primary peak of dimer particles that formed from +1/-1 charged particles is identified by its mobility diameter of ~1.26$D_{mono}$. Near the peak of the distribution, interference from particles other than +1/-1 dimers is minimal (c.f. Figure 7 in Petters, 2018).*

How does the RH affect the light scattering calculation?

> **Response:** All experiments were performed under dry conditions. Thus an RH correction was not required.

*Manuscript:* no changes were made to the text.

Considering the complex light scattering responding curve, please estimate the uncertainty caused by the multiple change and RH in your results, such as in fig. 4. Additional information should be provided to quantify the sizing capability of the modified POPS.

**Response:** The sizing capability of the modified POPS is in principle identical to the original POPS, but with a smaller range when using the MCA card. Uncertainties due to RH and multiple charges are highly application dependent and are appropriately discussed in the context of applications where such corrections are needed. The following changes were made to the manuscript.

*Manuscript: Atomized particles were dried to < 25% relative humidity (RH) using 4 or 5 silica-gel driers in series. At RH < 25% the water content for organic particles such as sucrose is negligible (Power et al., 2013). Ammonium sulfate is effloresced at RH < 40% and thus contains no water.*

Specific comments:

Line 63, What time and bin resolution do the DMA-OPC experiments need? How does the 10 Hz data improve the result compared with the 1 Hz data?

**Response Line 63:** Precise requirements will depend on the application. Based on the results shown in Figure 5 of the revised manuscript, a resolution of 500 bins is sufficient to resolve the transfer function of the DMA, where the mobility diameter is translated into an optical signal. The MCA data is useful here since it doesn't saturate at high concentration. Ten Hz data acquisition in DMA-OPC experiments may be useful to resolve concentration changes for fast mobility scans. However, the exact scan rate where this benefit will occur was not the focus of our work and needs to be determined in future studies.

Line 97, What is the uncertainty caused by the sheath flow variation?

**Response Line 97:** The purpose of the sheath flow is to prevent the optics from accumulating particles. We believe that it is small/negligible. First, the sheath flow is also not actively controlled in the regular POPS. It is simply taken from the ambient line. Second, concentration is determined from the sample flow rate, and thus does not depend on the sheath flow. Third, concentrations between the POPS and a CPC agree for sufficiently large sizes. Finally, we did test the setup with sample flow (not reported here), and found no change in concentration or sizing performance.

Figure 2, At what locations does the author monitor the phase transition temperature? Will the particle temperature decrease when transferring from SMPS to POPS?

**Response Figure 2:** The following was added to the manuscript.

*Manuscript: ... particles are passed through a coalescence chamber which exposes the particles for a fixed time to elevated temperature conditions, which is measured at the outside of the metal chamber. Upon exiting the chamber, the sample is cooled to ambient conditions. This arrests the*

*sintering process, allowing for the observation of partially coalesced particles. If the temperature is sufficiently high, the particles coalesce and transform from a touching bisphere dimer particle into a spherical particle.*

Line 155, Can the author distinguish the multiple charged particles with a POPS? If so, what is the size resolution of the modified POPS?

> **Response Line 155:** The multiple charges can be distinguished with POPS, as is discussed in Section 3.1. We estimate that relative changes in scattering intensity of ~2-3% can be resolved. The corresponding size resolution will depend on the refractive index, particle shape, and particle size/mass. The sizing capability of the modified POPS is in principle identical to the original POPS, but with a smaller range when using the MCA card.

*Manuscript: no changes were made to the text.*

Line 157, Do we expect the refractive index to change during this particle transition? If so, how can it affect the mode diameter determination?

> **Response Line 157:** We do not know if the refractive changes during the particle phase transition. However, the mode diameter is determined from the count vs. mobility distribution, like it is done for standard scanning mobility particle sizer measurements. Therefore, in this context the refractive index is not important.

Line 160, The viscosity of what chemical compound? Under what temperature?

Can the author quantify the enhancement?

**Response Line 160: Manuscript:** *The mapping between transition temperature and viscosity depends on particle size, surface tension, and residence time in the coalescence chamber (Rothfuss and Petters, 2016). The effect of surface tension on the mapping is small (Marsh et al, 2018). Here, the transition temperature corresponds to the condition when viscosity is ~$10^7$ Pa s.*

Line 265, does it suggest that the MCA modification narrows the size detection range of POPS? If it is true, that will limit the application of POPS. Is there any solution for that?

> **Response Line 265:** The MCA addition is not limiting the POPS application as the POPS serial and Digitizer-PH are able to capture lower and higher size detections while the MCA is attached (see Figure 5). It is in principle possible to use the MCA card output to measure the concentration and scale the Digitizer-PH size distribution accordingly. This requires the assumption that the coincidence only minimally affects the size distribution and the assumption that the concentration dynamics of < 200 nm particle is the same as for > 200 nm particles. This is also mentioned in the manuscript:

**Manuscript:** *Importantly, the addition of the MCA still preserves the Digitizer-PH output. Thus, the combination of the two may be used to reconstruct the size distribution from the Digitizer-PH using post-processing while scaling the concentration using the measured MCA-PH data.*

Fig 8. Does the author have a plot of the temperature scans of MCA-PH response? Is it different or similar to digitizer-PH?

> **Response Fig 8:** The selected particle size was too small to resolve changes in the MCA-PH response.

The nominal 166 nm is based on the electrical mobility size, right? If so, what is the corresponding optical size of the particles? Line 275, at 72 C, the mobility size is around 210 nm, similar to the doubly charged particles. How does the author separate the coalescence and the doubly-charged particles?

> **Response:** We do not explicitly map mobility and optical size in this paper as this mapping is sensitive to the refractive index. Based on the calibration shown in Figure 5, this size is at the limit of detection of the instrument. Particles with 166 mobility diameter are also charged and filtered by the electrostatic precipitator. The size of doubly charged particles is 261 nm, which is much larger than the dimer size. Furthermore, doubly charged particles transmitted by the size-selecting DMAs carry charge and thus are also filtered by the electrostatic precipitator. The following is added to the manuscript to refer the reader for further details:

**Manuscript:** *Note that the mobility distribution of the dual tandem DMA is complex and contains multiple modes. Details are given in Petters (2018) and Rothfuss et al. (2019). The primary peak of dimer particles that formed from +1/-1 charged particles is identified by its mobility diameter of ~$1.26D_{mono}$. Near the peak of the distribution, interference from particles other than +1/-1 dimers is minimal (c.f. Figure 7 in Petters, 2018).*

---

## Author Comment (AC2)

**08/19/2022**

**RC2: 'Comment on amt-2022-185', Anonymous Referee #2, 20 Jul 2022**

Author statement: The authors thank the referee for reviewing this manuscript. An itemized **response** for the rebuttal can be found below for each response given in blue. The tracked-changes version can be found below for each response in red and the already existing text is in *italic*.

General comments:

Authors have modified the commercial version of POPs by adding an external MCA card and changing the flow path, which have extended the use of POPs in laboratory studies and field measurements. For example, in tandem with DMA, and real-time processing of signals allows the high time resolution measurements of aerosols under high aerosol concentration conditions. The authors have detailed characteristics of concentration response, size response, time response of the modified one and compared them with the commercial one, and also showed an application example for phase state change detection. Overall, this is a good technique paper that facilitate the use of POPs, thus only have some minor comments.

Suggest authors give a picture of the modified version in Fig.1.

> **Response 1:** We added a picture of the modified version of the POPS.

[Figure]

1: MCA card (Amptek 8000D)
2: Power supply (LRS 75 12 Mean well)
3: POPS
4: 50 Ω terminator
5: Needle valve
6: Data acquisition board
7: Inlet flow
8: Vacuum pump line
9: Serial cable/serial port

*Figure 2:* *Photograph of the modified POPS.*

Suggest that Sect **2.2**, **2.3** and 2.4 should be merged correspondingly with Sect **3.1**, 3.2 **3.3**.

**Response 2:** Thank you for the suggestion. However, per standard convention, we prefer to keep Methods and Results as separate sections.

The commercial POPS is attractive for its relatively low cost, lightweight, low power consumption, thus very popular in characterizing vertical distributions of aerosols such as in unmanned aerial vehicle (UAV) measurements, does the light weight and power consumption of the modified version remain low? It might be better that authors add this information in discussion part, and discuss the potential usage of modified version in for example UAV measurements.

**Response 3: Manuscript:** *As shown in Fig. 2, the unit was placed inside a 19"* *rackmount enclosure. No effort was made to minimize the height and weight of the enclosure or the power consumption of the unit. The additional power requirement beyond the factory design depends on the vacuum pump. We used an ~400W model, but smaller pumps will be sufficient. The rackmount form factor is suitable for laboratory and field applications, including measurements from mobile platforms such as vehicles or airplanes where size and weight are less critical than in balloon-borne deployments. The cost to modify the factory supplied unit was ~US $6,500, including the MCA card, enclosure, vacuum pump, needle valve, power supply, Swagelok fittings, electrical connections, and particle filters.*

Specific comments:

L15 "The 90/10 rise and fall….." this sentence is too technical, it is hard to follow for readers without clear clarifications, I understand clearly what authors want to deliver only when I read L120 to 124.

**Response L15:** The text was revised as follows.

**Manuscript:** *The time required to change the concentration between 90-10 % and vice versa for a step change in concentration were measured to be 0.17 s and 0.41 s at a flow rate of 5 $cm^3$ $s^{-1}$*

L182 The given typical refractive index of organics should have a larger range (Moise et al., 2015)

**Response 182:** Thank you for bringing this information to our attention. At 405 nm wavelength, the real part of the refractive index of secondary organic aerosols range is between 1.4-1.7 (e.g. naphthalene is 1.58-1.66 at 405 nm).

**Manuscript:** *Typically, organics have a refractive index of 1.4-1.7 (Flores et al., 2014; He et al., 2018; Nakayama et al., 2012; Moise et at., 2015).*

Moise, T., Flores, J. M., and Rudich, Y.: Optical Properties of Secondary Organic Aerosols and Their Changes by Chemical Processes, Chemical Reviews, 115, 4400-4439, 10.1021/cr5005259, 2015.

---

## Author Comment (AC3)

**08/19/2022**

**RC3: 'Comment on amt-2022-185', Anonymous Referee #3, 21 Jul 2022**

Author statement: The authors thank the referee for reviewing this manuscript. An itemized **response** for the rebuttal can be found below for each response given in blue. The tracked-changes version can be found below for each response in red and the already existing text is in *italic*.

General comments:

A modified version of Commercial POPS is presented. Authors used a high-precision multichannel analyzer to improve count rate limitation. A needle valve and vacuum pump are used to provide additional flow stability. The overall goal of the manuscript is to characterize the modified POPS instrument along with the comparison with commercial one. Authors also presented a practical application of the modified POPS, which is the detection of phase transition using high-resolution pulse height data. The reviewer finds the manuscript exciting and a good read. Authors have taken care of explaining minute details and nicely present the work. Review recommends publications with the following comments/suggestions:

1. Reviewer is interested in knowing the commercial aspects of modified POPS, for example, cost, size, and portability. Can we still use the modified one for field measurement?

**Manuscript:** *As shown in Fig. 2, the unit was placed inside a 19" rackmount enclosure. No effort was made to minimize the height and weight of the enclosure or the power consumption of the unit. The additional power requirement beyond the factory design depends on the vacuum pump. We used an ~400W model, but smaller pumps will be sufficient. The rackmount form factor is suitable for laboratory and field applications, including measurements from mobile platforms such as vehicles or airplanes where size and weight are less critical than in balloon-borne deployments. The cost to modify the factory supplied unit was ~US $6,500, including the MCA card, enclosure, vacuum pump, needle valve, power supply, Swagelok fittings, electrical connections, and particle filters*.

2. Reviewer would suggest showing a laboratory picture of modified POPS setup/instrument.

   **Response 2:** We added a picture of the modified version of the POPS.

[Figure]

1: MCA card (Amptek 8000D)
2: Power supply (LRS 75 12 Mean well)
3: POPS
4: 50 Ω terminator
5: Needle valve
6: Data acquisition board
7: Inlet flow
8: Vacuum pump line
9: Serial cable/serial port

*Figure 2: Photograph of the modified POPS.*

3. Reviewer thinks section 2 should contain the details of the modified POPS, and sections like 2.1 to 2.4 should be discussed in the result section (3). Additionally, section 2.5 should go in the supplementary material.

   **Response 3:** Thank you for the suggestion. However, per standard convention, we prefer to keep Methods and Results as separate sections.

Minor comments:

1. Reviewer wonders why figures in the manuscript have a faded axis. Also, authors may wish to remove grid lines from figures.

   **Response 1:** We increased the darkness of the gridlines in the revised manuscript. We prefer to keep this style of plotting the data.

2. In figure 4, reviewer thinks it is better to give the plot names instead of indicating the top, bottom, and middle. The inset of figure 7a is not clearly visible to the reviewer, and a box should be added to indicate the ROI.

   **Response 2:** We added labels (a), (b), and (c) to Figure 4 and modified the caption.

**Manuscript:** *Figure 5: (a) Digitizer-PH and (b) MCA-PH histograms for dry size selected ammonium sulfate particles between 200 nm and 700 nm. The histograms were normalized such that the largest value equals unity. (c) Comparison of Digitizer-PH and MCA-PH at the peak of the histogram. Error bars correspond to the 95% confidence interval of the mode determined via a curve fit of the data to a lognormal distribution. The line corresponds to linear regression with the equation Digitizer-PH=MCA-PH\*50.59-92.57.*

3. L75, reviewer thinks author should write a line about how and why the 50 Ω terminator reduces the noise. Reviewer understands the use of a preamplifier, but what is the role of a shaping amplifier.

   **Response 3:** Significant noise was observed in the lower channels of the MCA. The 50 ohm terminator helped to reduce the noise. The shaping amplifier is part of the MCA card. As described in the documentation of the MCA card, the role of the shaping amplifier is to filter noise, to stabilize the baseline, and to provide enough gain for accurate measurement.

**Manuscript:** *A 50 Ω terminator was integrated to reduce electronic noise that was present in the lower MCA channels. The MCA card consists of a preamplifier and a shaping amplifier. The role of the shaping amplifier is to filter noise, to stabilize the baseline, and to provide enough gain for accurate measurement.*

4. L75, maybe reviewer misunderstood why high gain is 0-1V and low gain is 0-10V, expecting the reverse.

   **Response 4:** There are two different input voltage scales in MCA 8000D which correspond to 0-1 and 0-10 V.

**Manuscript:** *The card allows configuration for user selectable input ranges of either 0-1 V or 0-10 V. In this work, the 0-1 V setting was used.*

5. L95, what authors mean by critical flow conditions. Is it a predefined flow condition or maximum flow, or optimum flow?

   **Response 5:** Clarified as below.

**Manuscript**: *...is replaced with a needle valve and vacuum pump, operated at critical flow conditions. Critical flow refers to conditions where the flow across the orifice reaches sonic velocity and further decreases in downstream pressure have no further effect on the volumetric flow rate.*

6. In figure 1, the definition of HFDMA should be included in the abbreviations.

   **Response 6:** Done.

7. L105, from the reader's point of view, a sentence should be added to explain custom DMA. Furthermore, figure 2 should be moved to page 5.

   **Response 7:** The DMA is a high-flow DMA column operated at 9 L min$^{-1}$ sheath flow.

**Manuscript:** *The response of the Digitizer-PH and MCA-PH data were characterized downstream of a DMA. A schematic of the setup is shown in Fig. 2. ...The DMA consisted of a high-flow column (Stolzenburg et al., 1998) operated at 9 L min$^{-1}$ sheath flow.*

8. L115, author compared the results of other bins? If a higher bin number is assumed to give better statistics, 1000 bins would be more appropriate.

   **Response 8:** The 500 bin resolution was a compromise between size resolution and counting statistics. As shown in Figure 4, 500 bins are sufficient to resolve the DMA distribution. Higher bin resolution will not improve the calibration accuracy.

9. L140, reviewer did not understand the meaning of a high-level overview. Is it a detailed overview?

   **Response 9:** Changed as follows:

**Manuscript:** *Figure 3 provides an overview of the setup.*

10. L140, reviewer thinks the short form of HFDMA should be introduced earlier. Maybe in figure 1 or in L105.

    **Response 10:** The abbreviation explanation is now also given in Figure 1 in the main manuscript.

11. L145, Form reader point of view unit should be consistent. Author can use either inches or m.

    **Response 11:** In this line we prefer to give both. Units should generally be given in metric but the imperial units here add context for why these values were selected.